# WHEN SCALING MEETS LLM FINETUNING: THE EFFECT OF DATA, MODEL AND FINETUNING METHOD

**Biao Zhang**[†]   **Zhongtao Liu**[◇]   **Colin Cherry**[◇]   **Orhan Firat**[†]
[†]Google DeepMind    [◇]Google Research
{biaojiaxing,zhongtao,colincherry,orhanf}@google.com

## ABSTRACT

While large language models (LLMs) often adopt *finetuning* to unlock their capabilities for downstream applications, our understanding on the inductive biases (especially the scaling properties) of different finetuning methods is still limited. To fill this gap, we conduct systematic experiments studying whether and how different scaling factors, including LLM model size, pretraining data size, new finetuning parameter size and finetuning data size, affect the finetuning performance. We consider two types of finetuning – full-model tuning (FMT) and parameter efficient tuning (PET, including prompt tuning and LoRA), and explore their scaling behaviors in the data-limited regime where the LLM model size substantially outweighs the finetuning data size. Based on two sets of pretrained bilingual LLMs from 1B to 16B and experiments on bilingual machine translation and multilingual summarization benchmarks, we find that 1) LLM finetuning follows a power-based multiplicative joint scaling law between finetuning data size and each other scaling factor; 2) LLM finetuning benefits more from LLM model scaling than pretraining data scaling, and PET parameter scaling is generally ineffective; and 3) the optimal finetuning method is highly task- and finetuning data-dependent. We hope our findings could shed light on understanding, selecting and developing LLM finetuning methods.

## 1 INTRODUCTION

Leveraging and transferring the knowledge encoded in large-scale pretrained models for downstream applications has become the standard paradigm underlying the recent success achieved in various domains (Devlin et al., 2019; Lewis et al., 2020; Raffel et al., 2020; Dosovitskiy et al., 2021; Baevski et al., 2020), with the remarkable milestone set by large language models (LLMs) that have yielded ground-breaking performance across language tasks (Brown et al., 2020; Zhang et al., 2022b; Scao et al., 2022; Touvron et al., 2023). Advanced LLMs, such as GPT-4 (OpenAI, 2023) and PaLM 2 (Anil et al., 2023), often show emergent capabilities and allow for in-context learning that could use just a few demonstration examples to perform complex reasoning and generation tasks (Wei et al., 2022; Zhang et al., 2023; Fu et al., 2023; Shen et al., 2023). Still, LLM finetuning is required and widely adopted to unlock new and robust capabilities for creative tasks, get the most for focused downstream tasks, and align its value with human preferences (Ouyang et al., 2022; Yang et al., 2023; Gong et al., 2023; Schick et al., 2023). This becomes more significant in traditional industrial applications due to the existence of large-scale annotated task-specific data accumulated over years.

There are many potential factors affecting the performance of LLM finetuning, including but not limited to 1) pretraining conditions, such as LLM model size and pretraining data size; and 2) finetuning conditions, such as downstream task, finetuning data size and finetuning methods. Intuitively, the pretraining controls the quality of the learned representation and knowledge in pretrained LLMs, and the finetuning affects the degree of transfer to the donwstream task. While previous studies have well explored the scaling for LLM pretraining or training from scratch (Kaplan et al., 2020; Hoffmann et al., 2022) and the development of advanced efficient finetuning methods (Hu et al., 2021; He et al., 2022), the question of whether and how LLM finetuning scales with the above factors unfortunately receives very little attention (Hernandez et al., 2021), which is the focus of our study.

Note, apart from improving finetuning performance, studying the scaling for LLM finetuning could help us to understand the impact of different pretraining factors from the perspective of finetuning, which may offer insights for LLM pretraining.

In this paper, we address the above question by systematically studying the scaling for two popular ways of LLM finetuning: *full-model tuning* (FMT) that updates all LLM parameters and *parameter-efficient tuning* (PET) that only optimizes a small amount of (newly added) parameters, such as prompt tuning (Lester et al., 2021, Prompt) and low-rank adaptation (Hu et al., 2021, LoRA). We first examine finetuning data scaling (Hernandez et al., 2021), on top of which we further explore its scaling relationship with other scaling factors, including LLM model size, pretraining data size, and PET parameter size. We focus on the data-limited regime, where the finetuning data is much smaller than the LLM model, better reflecting the situation in the era of LLM. For experiments, we pretrained two sets of bilingual LLMs (English&German, English&Chinese) with model size ranging from 1B to 16B, and performed large-scale study on WMT machine translation (English-German, English-Chinese) and multilingual summarization (English, German, French and Spanish) tasks with up to 20M finetuning examples. Our main findings are summarized below:

- We propose the following multiplicative joint scaling law for LLM finetuning:

$$\hat{\mathcal{L}}(X, D_f) = A * \frac{1}{X^\alpha} * \frac{1}{D_f^\beta} + E,  \tag{1}$$

  where $\{A, E, \alpha, \beta\}$ are data-specific parameters to be fitted, $D_f$ denotes finetuning data size, and $X$ refer to each of the other scaling factors. We show empirical evidence that this joint law generalizes to different settings.

- Scaling LLM model benefits LLM finetuning more than scaling pretraining data.

- Increasing PET parameters doesn't scale well for LoRA and Prompt, although LoRA shows better training stability.

- The scaling property for LLM finetuning is highly task- and data-dependent, making the selection of optimal finetuning method for a downstream task non-trivial.

- LLM-based finetuning could encourage zero-shot generalization to relevant tasks, and PET performs much better than FMT.

## 2 SETUP

**Downstream Tasks**  We consider machine translation and multilingual summarization as the downstream tasks for the finetuning, because 1) these tasks require resolving cross-lingual under-standing and generation, which represent high complexity and are challenging; and 2) they are well established in NLP with rich amount of available finetuning corpora. Specially, we adopt WMT14 English-German (En-De) and WMT19 English-Chinese (En-Zh) (Kocmi et al., 2022) for transla-tion. We combine the De, Spanish (Es) and French (Fr) portion of the multilingual summarization dataset (Scialom et al., 2020) with CNN/Daily-Mail (Hermann et al., 2015, En) for summarization and denote it as MLSum. Details about each task are listed in Table 1a. Note for MLSum, we di-rectly concatenate the datasets of different languages for training and evaluation, where each article is prepended a prompt indicating its language "*Summarize the following document in {lang}:*".

**LLMs and Preraining**  We adopt the exact setup as in Garcia et al. (2023) for LLM pretraining. The model is a decoder-only Transformer with multi-query attention (Chowdhery et al., 2022) and trained with the modified UL2 objective (Tay et al., 2022). Considering the focused downstream tasks and also to ensure the generalization of our study, we pretrained two sets of bilingual LLMs, i.e. En-De LLM and En-Zh LLM. The pretraining data is a mix of monolingual data from two languages: we use En/De (En/Zh) data with about 280B (206B) tokens to pretrain the En-De (En-Zh) LLM. We train LLMs with parameter sizes from 1B to 16B by varying model configurations as in Table 3 and keep all other settings intact. All LLMs are optimized using Adafactor (Shazeer & Stern, 2018) for one training epoch under a cosine learning rate decay schedule (from 0.01 to 0.001). We refer the readers to (Garcia et al., 2023) for more details about the pretraining.

Table 1: Setups for finetuning. "K/B/M": thousand/billion/million; "#Train": the number of training examples; "Length": maximum source/target sequence length cut at training. Note pretraining data size is for token count. **Bold** numbers denote the held-out settings we leave for scaling law verification.

(a) Details for finetuning tasks.

| Task | #Train | Length | Dev | Test | Zero-Shot | Base LLM |
|------|--------|--------|-----|------|-----------|----------|
| WMT14 En-De | 4.5M | 256/256 | newstest2013 | newstest2020,2021,2022 | Flores200 | En-De LLM |
| WMT19 En-Zh | 25M | 256/256 | newsdev2017 | newstest2020,2021,2022 | Flores200 | En-Zh LLM |
| MLSum | 1.1M | 512/256 | official dev sets | official test sets | - | En-De LLM |

(b) Scaling settings for different factors.

| | | |
|---|---|---|
| LLM Model Sizes | | 1B, 2B, 4B, 8B, **16B** |
| Pretraining Data Sizes | En-De LLM | 84B, 126B, 167B, **209B**, 283B |
| | En-Zh LLM | 84B, 105B, 126B, 147B, **167B**, 206B |
| PET Parameter Sizes | Prompt Length | 50, 100, 150, 200, 300, 400, **600** |
| | LoRA Rank | 4, 8, 16, 32, 48, 64, **128** |
| Finetuning Data Sizes | Prompt & LoRA | 8K, 10K, 20K, 30K, 40K, 50K, 60K, 70K, 80K, 90K, **100K** |
| | FMT– WMT En-De | 100K, 500K, 1M, 1.5M, 2M, 2.5M, 3M, 3.5M, 4M, **4.5M** |
| | FMT– WMT En-Zh | 1M, 2M, 3M, 4M, 5M, 10M, 15M, 20M, **25M** |
| | FMT– MLSum | 100K, 200K, 300K, 400K, 500K, 600K, 700K, 800K, **900K** |

**Finetuning Settings** We mainly study the scaling for the following three finetuning methods:

- **Full-Model Tuning (FMT)**: This is the vanilla way of finetuning which simply optimizes all LLM parameters;

- **Prompt Tuning (Prompt)**: Prompt prepends the input embedding $X \in \mathbb{R}^{|X| \times d}$ with a tunable "soft-prompt" $P \in \mathbb{R}^{|P| \times d}$, and feeds their concatenation $[P; X] \in \mathbb{R}^{(|P|+|X|) \times d}$ to LLM. $|\cdot|$ and $d$ denote sequence length and model dimension, respectively. During finetuning, only the prompt parameter $P$ is optimized. We initialize $P$ from sampled vocabulary, and set the prompt length $|P|$ to 100 by default (Lester et al., 2021).

- **Low-Rank Adaptation (LoRA)**: Rather than modifying LLM inputs, LoRA updates pretrained model weights $W \in \mathbb{R}^{m \times n}$ with trainable pairs of rank decomposition matrices $B \in \mathbb{R}^{m \times r}, A \in \mathbb{R}^{r \times n}$, and uses $W + BA$ instead during finetuning. $m, n$ are dimensions and $r$ is LoRA rank. Only $B$s and $A$s are optimized. We apply LoRA to both attention and feed-forward layers in LLMs, and set the rank $r$ to 4 by default (Hu et al., 2021).

We explore 4 different factors for the scaling, which are summarized in Table 1b. Except LLM model scaling, all experiments are based on the corresponding 1B LLM. For pretraining data scaling, we adopt intermediate pretrained checkpoints as the proxy due to computational budget constraint while acknowledge its sub-optimality. Details for optimization are given in Appendix.

**Evaluation** We use the best checkpoint based on token-level perplexity (PPL) on the dev set for evaluation. For scaling laws, we report PPL on test sets; for general generation, we use greedy decoding, and report BLEURT (Sellam et al., 2020) and RougeL (Lin, 2004) for translation and summarization, respectively. For zero-shot evaluation, we adopt Flores200 (NLLB Team, 2022) and evaluate on {Fr, De, Hindi (Hi), Turkish (Tr), Polish (Po)→Zh} and {Fr, Zh, Hi, Tr, Po→De} for En-Zh and En-De translation respectively. For scaling law evaluation, we split empirical data points into two sets, *empirical fitting* and *held-out* set, where the former is used for fitting scaling parameters and the latter is used for evaluation. We report mean absolute deviation. To reduce noise, we perform three runs, each with a different random subset of the finetuning data, and report average performance. When sampling for MLSum, we keep the mixing ratio over different languages fixed.

## 3 WHY MULTIPLICATIVE JOINT SCALING LAW?

We consider 4 scaling factors in this study but jointly modeling all of them is time and resource consuming. Instead, we treat finetuning data as the pivoting factor and perform joint scaling analysis

Figure 1: Fitted single-variable scaling laws for finetuning data scaling over different LLM model sizes on WMT14 En-De. Solid lines denote fitted scaling curves. Filled circles and triangles denote fitting and held-out data points. $\Delta_h$: mean absolute deviation on the held-out data.

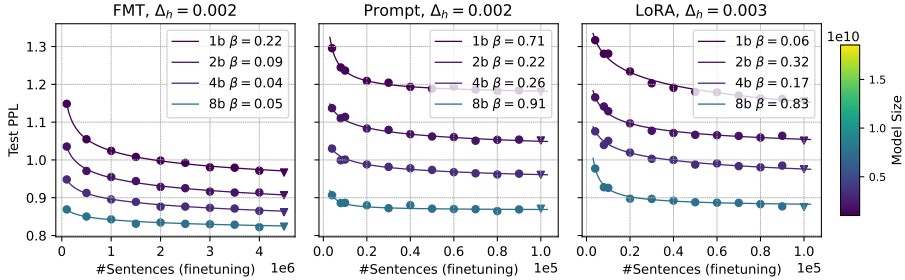

Table 2: Held-out fitting errors ($\downarrow$) for the additive and multiplicative scaling formulation over different finetuning methods on WMT14 En-De. Multiplicative scaling law generalizes better.

| Scaling Factor | Multiplicative | | | | Additive | | | |
|---|---|---|---|---|---|---|---|---|
| | FMT | Prompt | LoRA | Avg | FMT | Prompt | LoRA | Avg |
| LLM Model Size | 0.0052 | 0.0043 | 0.0047 | **0.0048** | 0.012 | 0.0076 | 0.0045 | 0.0079 |
| Pretraining Data Size | 0.0057 | 0.0061 | 0.0084 | **0.0068** | 0.0048 | 0.0075 | 0.0082 | 0.0069 |
| PET parameter size | - | 0.005 | 0.0031 | **0.004** | - | 0.0069 | 0.0032 | 0.005 |

between it and every other factor separately. Below, we start with finetuning experiments for FMT, Prompt and LoRA on WMT14 En-De, and then explore the formulation for the joint scaling.

**Finetuning data scaling follows a power law.** We first examine the scaling over finetuning data size for each LLM model size independently, with a single variable formulation: $\hat{\mathcal{L}}(D_f) = A/D_f^\beta + E$. Following Hoffmann et al. (2022), we estimate $\{A, \beta, E\}$ using the Huber loss ($\delta = 0.001$) and the L-BFGS algorithm, and select the best fit from a grid of initializations. Figure 1 shows that the above formulation well describes LLM finetuning data scaling with small predictive errors across model sizes and methods, echoing with the findings of Hernandez et al. (2021). Such scaling trend also implies that, while finetuning with small amount of examples could achieve decent results (Zhou et al., 2023; Gao et al., 2023), larger scale finetuning data still contributes to improved downstream performance, especially when the downstream application is well defined.

**Additive or multiplicative joint scaling law for LLM finetuning?** Figure 1 also shows some scaling pattern over LLM model sizes, suggesting the existence of a joint scaling law. We explore two formulations: *multiplicative* as in Eq. (1) and *additive*: $\hat{\mathcal{L}}(X, D_f) = A/X^\alpha + B/D_f^\beta + E$ (Hoffmann et al., 2022), and compare them via empirical experiments.[1]

In both formulations, $\alpha$ and $\beta$ reflect the impact of factor $X$ and finetuning data size on the performance, respectively, which are factor-specific. $E$ is a model- and task-dependent term, describing irreducible loss (Ghorbani et al., 2021). We notice that the meaning for $\beta$ and $E$ generalizes over different factors $X$, and thus propose to estimate them first based on results for both LLM model and pretraining data scaling.[2] Such joint fitting could also reduce overfitting and improve extrapolation ability. We apply the following joint fitting loss:

$$\min_{a_X, b_X, \alpha_X, \beta, e} \sum_{run\ i\ in\ factor\ X} \text{Huber}_\delta \left( \hat{\mathcal{L}}\left(X^i, D_f^i | a_X, b_X, \alpha_X, \beta, e\right) - \mathcal{L}^i \right), \tag{2}$$

---

[1]For LLM model scaling, we omitted the newly added parameters in PET because 1) the added parameters only take a very tiny proportion, and 2) the proportion across LLM model sizes is similar. Take the 1B LLM as example. $|P| = 100$ in Prompt adds 0.017% parameters; $r = 4$ in LoRA adds 0.19% parameters. We also explored different formulations for the new parameters for PET, which don't make a substantial difference.

[2]We didn't consider PET parameter scaling when estimating $\beta$ and $E$ because this scaling is pretty weak and ineffective, as shown in Section 4.

Figure 2: Fitted multiplicative joint scaling laws for **LLM model size and finetuning data size** on WMT14 En-De, WMT19 En-Zh and MLSum. $\Delta_e/\Delta_h$: mean absolute deviation on the fitting/held-out data. $\alpha_m/beta$: scaling exponent for LLM model size/finetuning data size. We work on 1B to 16B LLM.

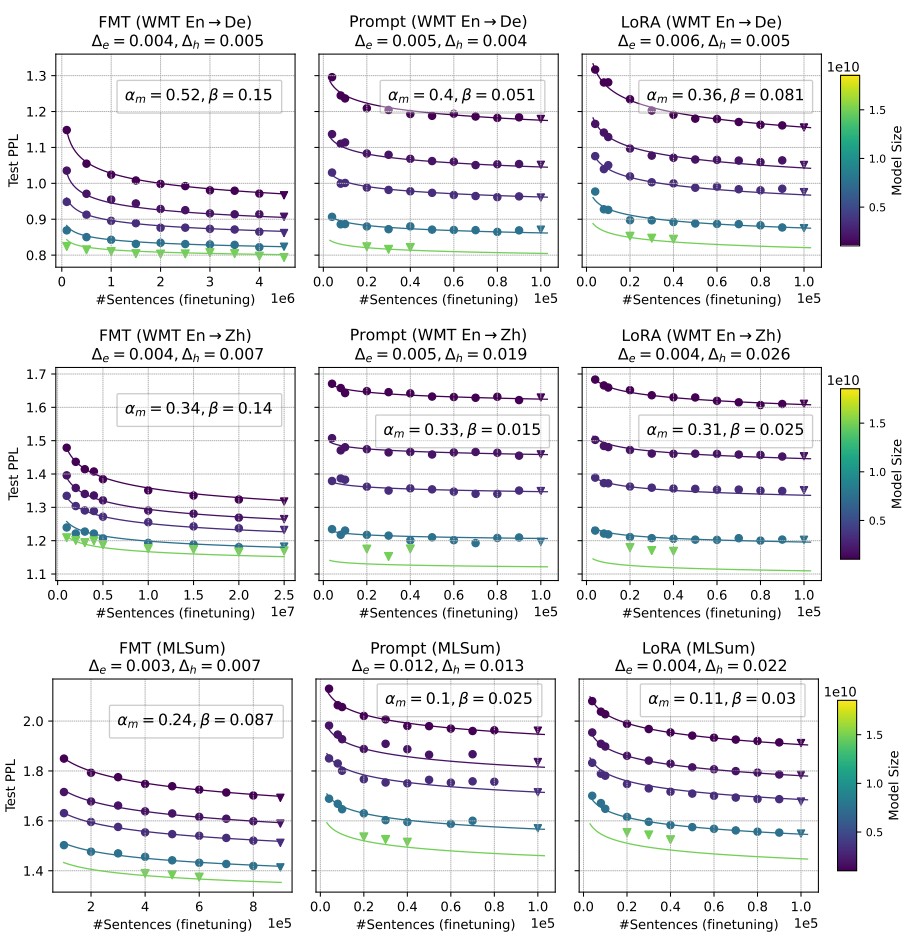

where we set $A_X = e^{ax}, B_X = e^{bx}, E = e^e$, and $X$ refers to LLM model size or pretraining data size. Note $b_X$ is only valid in the additive formulation. We then fix $\beta$ and $E$ and refit other parameters for each factor, separately.

Table 2 (and Table 6 in Appendix) shows that both joint laws perform similarly while the multiplicative one achieves slightly lower extrapolation error on average. Therefore, we adopt Eq. (1) for follow-up analysis.

## 4   SCALING RESULTS FOR LLM FINETUNING

Here, we show the empirical results for LLM model, pretraining data and PET parameter scaling on WMT14 En-De, WMT19 En-Zh and MLSum in Figures 2, 3 and 4, respectively. Results for BLEURT/RougeL are given in Appendix (Figures 7, 8 and 9), which shows high correlation with the PPL scores in general (see Table 7). Fitted scaling parameters are summarized in Table 4.

**The proposed multiplicative scaling law captures the scaling relation between different factors and finetuning data size.** In each group of experiments, we leave several data points along each scaling dimension as the held-out set. We report the mean absolute derivation on the empirical fitting ($\Delta_e$) and held-out ($\Delta_h$) sets to show the fitting and predictive ability, respectively. In general, we observe that Eq. (1) captures the scaling trend of different factors under finetuning data scaling with small fitting and extrapolation errors. Note there are some mismatched cases, where the empirical data points themselves could be noisy mostly caused by unstable optimization and

Figure 3: Fitted multiplicative joint scaling laws for **pretraining data size and finetuning data size** on WMT14 En-De, WMT19 En-Zh and MLSum(LLM model size: 1B). $\alpha_p$: scaling exponent for pretraining data size.

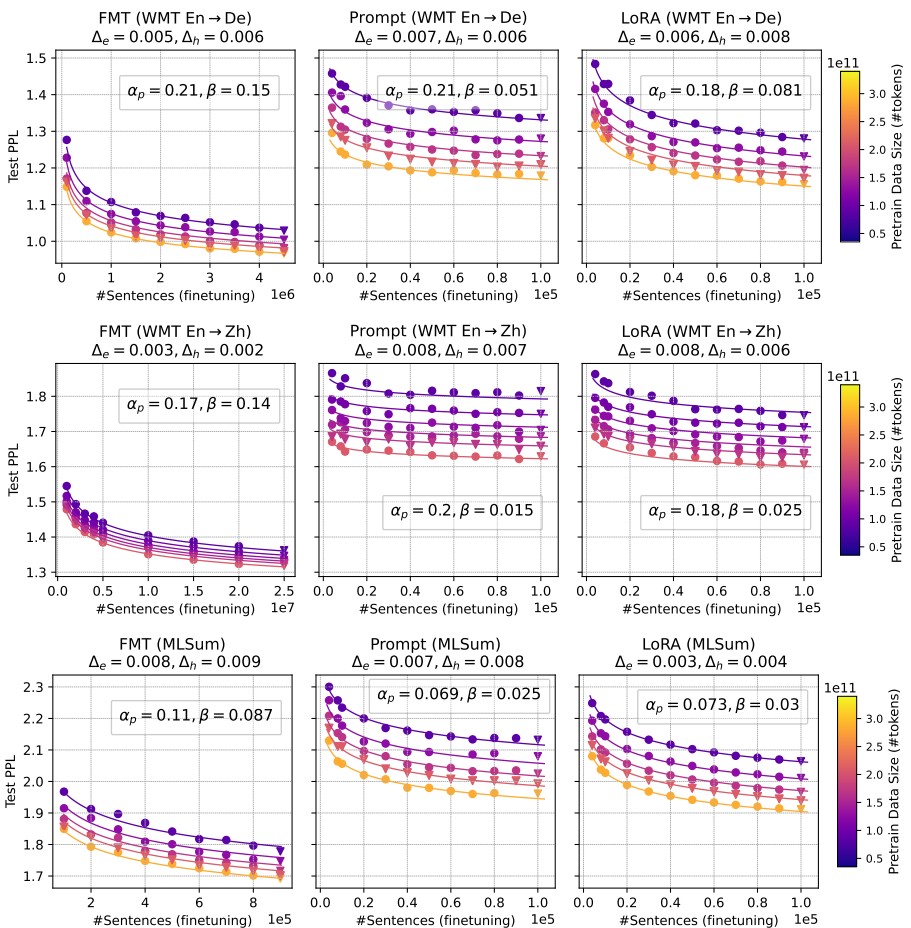

dev-set overfitting, challenging issues when tuning on small datasets. We observe high mismatch when extrapolating to 16B, particularly for LoRA and Prompt on WMT19 En-Zh in Figure 2. We ascribe this to 1) the insufficiency of empirical data over LLM model sizes (i.e. only 4 points) – the prediction by the fitted scaling law makes sense intuitively based on 1B-8B results, and 2) the inferior of the 16B En-Zh LLM due to pretraining instability, where its pretraining performance is not well predicted by even single-variable scaling laws as in Figure 10, Appendix.

**LLM finetuning benefits more from LLM model scaling than pretraining data scaling across tasks and methods.** While LLM model size and pretraining data size show similar impact on the pretraining scaling following the optimal scaling under a computational budget constraint (Hoffmann et al., 2022; Muennighoff et al., 2023), they show slightly different roles in finetuning scaling. Intuitively, finetuning heavily relies on the knowledge encoded in the LLM, where LLM model size and pretraining data size both matter. However, results in Figures 2, 3 and Table 4 show that the scaling exponent for LLM model size $\alpha_m$ often outnumbers that for pretraining data size $\alpha_p$ across finetuning methods and tasks, i.e. $\alpha_m > \alpha_p$. This suggests that using a larger LLM model is preferred over pretraining on a larger dataset, but we also notice that the difference in scaling is highly task-dependent. Our selection of closed generation tasks, i.e. translation and summarization, might deliver biased observations and for more creative generation tasks, larger and diverse pretraining data could be more crucial.

**Scaling PET parameters is ineffective, delivering limited gains for both LoRA and Prompt.** The amount of newly added trainable parameters often forms a bottleneck for the expressivity of

Figure 4: Fitted multiplicative joint scaling laws for **PET parameter size and finetuning data size** on WMT14 En-De, WMT19 En-Zh and MLSum(LLM model size: 1B). $\alpha_t$: scaling exponent for PET parameter size.

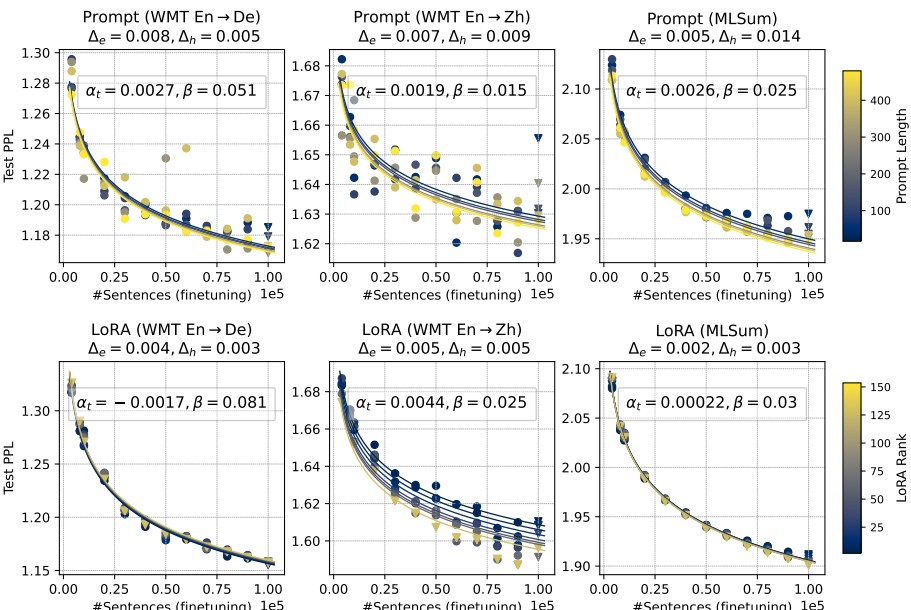

PET, controlled by the length $|P|$ and rank $r$ in Prompt and LoRA, respectively. However, Figure 4 and Table 4 show that increasing PET parameter sizes (i.e. enlarging $|P|$ and $r$) affects finetuning performance marginally as demonstrated by the small scaling exponents, $|\alpha_t| \ll 1e-2$, and even results in inverse scaling in some settings, e.g. LoRA on En-De. Besides, we observe that scaling Prompt length suffers from training instability as optimizing larger prompt embedding becomes non-trivial, which has also been seen in previous studies (Lester et al., 2021; Hu et al., 2021). We expect that carefully optimizing finetuning hyperparameters and prompt initialization may alleviate it to some extent. In this respect, LoRA is more stable and reliable.

**Finetuning data have more pronounced influence on FMT than PET, where LoRA scales better than Prompt.** Different finetuning methods show different degrees of finetuning data scaling. Table 4 shows that the scaling exponent $\beta$ for FMT is often significantly higher than that for PET across settings, indicating that FMT is more data-hungry and also benefits more from increasing finetuning data. While the scaling exponents are quite similar across PET, $\beta$ for LoRA often slightly surpasses that for Prompt. As shown in Figures 2, 3 and 4, LoRA often achieves better finetuning performance with more finetuning data than Prompt while Prompt behaves better with only few thousands of finetuning examples.

**PET depends more on LLM model and pretraining data scaling than finetuning data scaling across settings.** Since the majority of LLM parameters is frozen during finetuning, PET relies heavily on the encoded knowledge in pretrained LLMs when adapting them to downstream tasks. This is reflected by Table 4 that $\alpha_m$ and $\alpha_p$ are clearly larger than $\beta$ in PET. Figure 2 and 3 further support the scaling of LLM model, where the performance gap between FMT and PET is substantially narrowed with larger LLMs.

## 5  DISCUSSION

**Which finetuning method should we apply for a given task?** Unfortunately, there is no universal answer! Intuitively, there exists a critical point for finetuning data size beyond which one finetuning method performs better than another. However, the high non-linearity of the joint scaling law hinders us from identifying such points analytically, although the finetuning data size follows a power law when the performance difference between two methods is fixed (see Appendix). We thus resort to

Figure 5: Critical finetuning data sizes between different finetuning methods estimated by the fitted joint scaling law on WMT14 En-De, WMT19 En-Zh and MLSum. We use *scipy.optimize.fsolve* for the estimation. Critical point for "A vs. B": the finetuning data size (y-axis) at which A performs equal to B under the base model condition at x-axis. The value varies greatly across tasks.

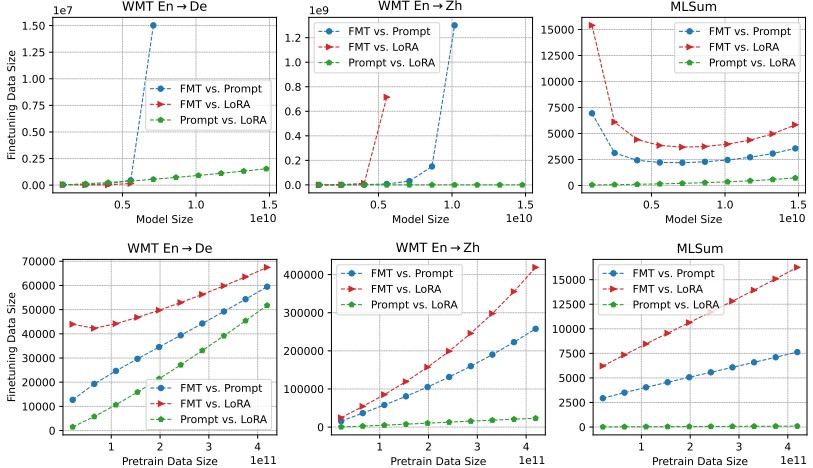

Figure 6: Zero-shot evaluation for LLM model size and finetuning data size scaling. The score is averaged over {Fr, De, Hi, Tr, Po→Zh} and {Fr, Zh, Hi, Tr, Po→De} for WMT19 En-Zh and WMT14 En-De, respectively.

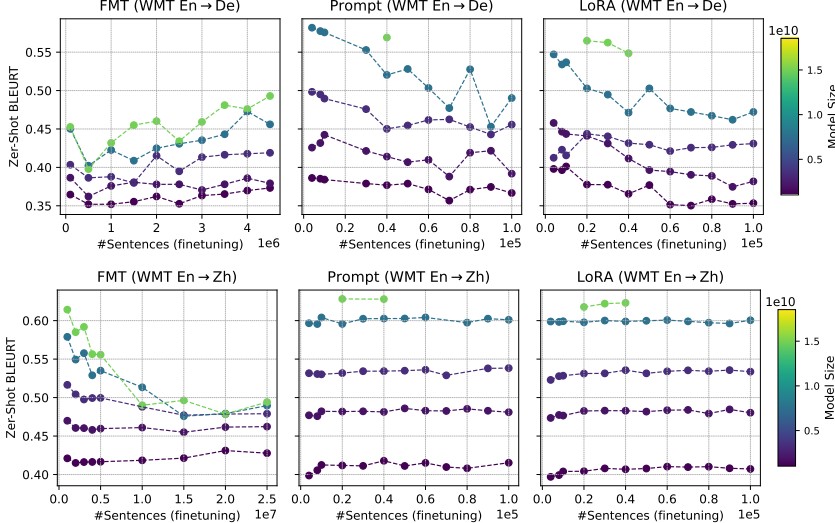

empirical methods by extrapolating the fitted scaling law. Figure 5 shows the critical points as a function of LLM model size and pretraining data size over different tasks.

The scaling trend and actual value are highly dependent on the downstream task: critical points for one task can hardly generalize to other tasks. Still, the existence of such points suggests that the selection of finetuning methods should be based on the availability of finetuning examples. When only few thousands of finetuning examples are available, PET should be considered first, either Prompt or LoRA. With sightly larger datasets, LoRA would be preferred due to its stability and slightly better finetuning data scalability. For million-scale datasets, FMT would be good.

**How does finetuning affect the generalization capability of the base LLM?** While finetuning on task-specific data improves task-specific performance, it may specialize the base LLM towards the task and hurt the models' generalization. We examine this for different finetuning methods by performing zero-shot translation for LLMs finetuned on WMT14 En-De and WMT19 En-Zh (Few-shot results are in Appendix). We focus on generalization to related tasks, where the target language is shared, i.e. De and Zh, and generalization should be relatively easier (Johnson et al., 2017). We report average performance for translation from a diverse set of source languages other than English.

Figure 6 shows the results. While specializing on a downstream task, finetuning could still elicit and improve the generalization for closely related tasks, although the overall zero-shot translation quality is inferior. Note whether finetuning benefits generalization is method- and task-dependent. Overall, Prompt and LoRA achieve relatively better results than FMT particularly when the base LLM is large, mostly because LLM parameters are frozen and the learned knowledge get inherited. This also suggests that when generalization capability is a big concern, PET should be considered.

## 6    RELATED WORK

**LLM finetuning**    With the significant increase of model size, updating all LLM parameters becomes computationally inefficient and unaffordable. Researchers thus resort to parameter efficient tuning methods that target achieving the best performance with minimal tunable parameters. Efforts in this direction mainly focus on developing efficient tunable modules for LLMs, such as adapters that insert small feed-forward layers (Houlsby et al., 2019; Bapna et al., 2019), prefix and prompt tuning that appends tunable embeddings to the input (Li & Liang, 2021; Lester et al., 2021), LoRA and compacter that adopts low-rank decomposition (Hu et al., 2021; Mahabadi et al., 2021), Bitfit that adds tunable bias vectors (Zaken et al., 2021), IA3 that scales model activations (Liu et al., 2022) and QLoRA that leverages quantization (Dettmers et al., 2023), to name a few. While previous studies reported encouraging performance with PET, e.g. reaching and even surpassing FMT across various domains (He et al., 2022; Ding et al., 2022; Liu et al., 2022; Dettmers et al., 2023), they mainly focus on one or few experimental setups, leaving the question of how scaling affects the performance of different finetuning methods under-explored.

**Scaling Laws**    Recent research has shown that the performance of neural models can be predicted by a power-law of model and/or data sizes (Hestness et al., 2017; Kaplan et al., 2020). Such pattern widely exists across different domains and model architectures, such as computer vision (Zhai et al., 2021), autoregressive generative modeling (Henighan et al., 2020), neural machine translation (Gordon et al., 2021; Ghorbani et al., 2021; Bansal et al., 2022; Zhang et al., 2022a), multilingual translation (Fernandes et al., 2023), multi-modal modeling (Aghajanyan et al., 2023) and sparse neural architectures (Frantar et al., 2023). These laws provide a valuable tool for guiding training decisions (Hoffmann et al., 2022) and model development by understanding how model performance evolves with scale, which greatly facilitates the development of LLMs (OpenAI, 2023). Unfortunately, the study of scaling for LLM finetuning lags behind badly, and our study fills this gap.

The most closely related work to ours is (Hernandez et al., 2021) which explored the scaling for knowledge transfer by comparing finetuning with training from scratch. Our study is orthogonal to theirs with significant difference as our key focus is understanding the scaling of different factors for LLM finetuning, rather than the transfer.

## 7    CONCLUSION AND FUTURE WORK

In this paper, we systematically studied the scaling for LLM finetuning, considering different factors including LLM model size, pretraining data size, finetuning data size, PET parameter size and diverse finetuning methods. To ensure the generality, we worked on two sets of LLMs, three different downstream tasks (translation and summarization), and three finetuning methods (FMT, Prompt and LoRA). We proposed a multiplicative joint scaling law that could describe the scaling relationship between finetuning data size and each other scaling factor. Extensive results show that increasing LLM model size has a higher impact on finetuning than pretraining data scaling, and that scaling PET parameter is ineffective. In addition, finetuning scaling is highly task- and data-dependent, making the selection of best finetuning method for a downstream task less conclusive.

We acknowledge that our work suffers from some limitations. The proposed joint scaling law is mostly based on empirical results on closed generation tasks without theoretical groundings. Whether it could generalize to different finetuning scenarios requires more experimentation, which however is beyond our current computing budget. Besides, we understand the imperfection of the optimization and evaluation for Prompt and LoRA in some setups. In the future, we would like to extend our study to multi-modal LLMs, explore the impact of finetuning data quality and consider open and creative generation tasks as well as multi-task setup for finetuning.

## 8    ACKNOWLEDGEMENTS

We thank the reviewers for their insightful comments. We thank Yamini Bansal for providing valuable feedback on the scaling laws, Xavier Garcia for reviewing this work with constructive comments, Frederick Liu for helpful discussion on PET optimization, and Quoc Le, Apu Shah and Google Translate team for supporting this research.

We also thank the colleagues building the training infrastructure used in this paper: Brian Lester, Rami Al-Rfou and Noah Constant for prompt tuning, Chu-Cheng Lin for LoRA, Xavier Garcia and the T5X team (Roberts et al., 2023) for the training framework.

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

## A  APPENDIX

Table 3:  Hyperparameters for different-sized LLMs. "B": billion; "#Layers, #Heads": the number of layers and attention heads, respectively; "Head Dim, FFN Dim, Model Dim": the dimension for each attention head, the feed-forward layer and the hidden representation, respectively.

| LLM Model Size | #Layers | #Heads | Head Dim | FFN Dim | Model Dim |
|---|---|---|---|---|---|
| 1B | 16 | 8 | 256 | 8192 | 2048 |
| 2B | 20 | 10 | 256 | 10240 | 2560 |
| 4B | 24 | 12 | 256 | 12288 | 3072 |
| 8B | 32 | 16 | 256 | 16384 | 4096 |
| 16B | 40 | 20 | 256 | 20480 | 5120 |

Figure 7:  Generation quality (BLEURT/RougeL) for scaling **LLM model size and finetuning data size** on WMT14 En-De, WMT19 En-Zh and MLSum. Overall, BLEURT/RougeL correlates positively with PPL with few exceptions.

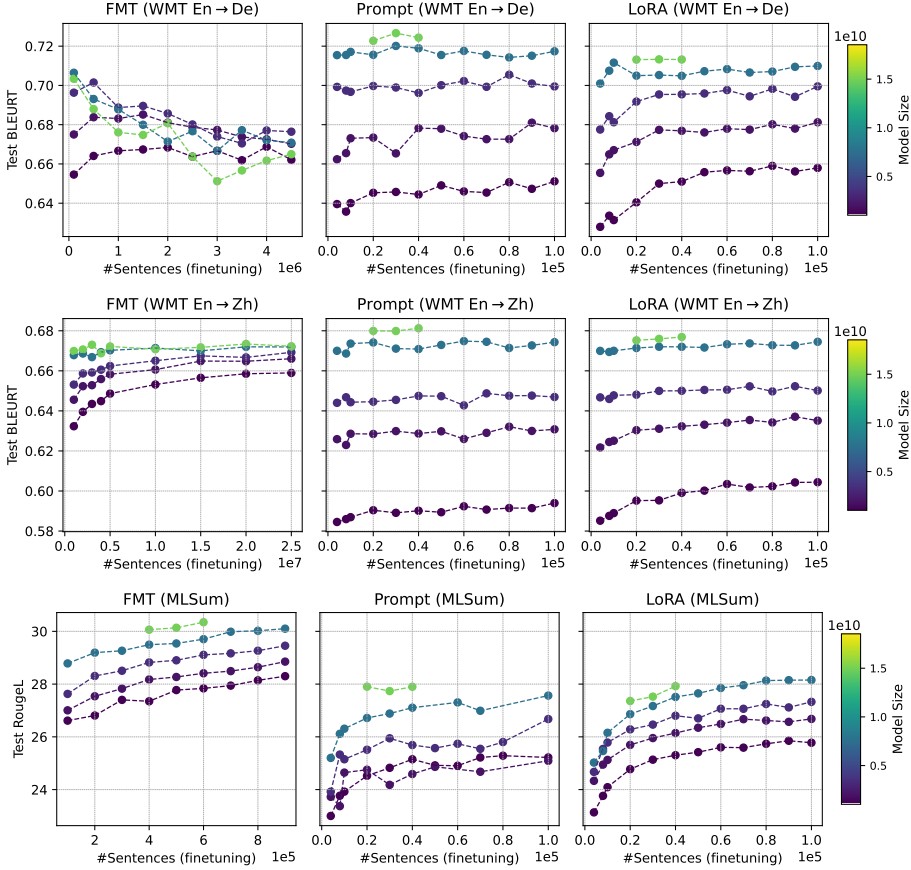

**Optimization for LLM finetuning.**    For optimization, we continue the pretraining from the given pretrained checkpoint on finetuning data but with the standard conditional log-likelihood loss. More specifically, for each finetuning example, we concatenate the *input* and *target* into a single sequence and compute the log-likelihood on the *target* alone. Adafactor and cosine learning rate schedule are reused. Note En-De and En-Zh LLM are pretrained for 135K and 98K steps, respectively. All LLMs are further finetuned for up to 200K steps (except for WMT En-Zh (FMT) which is 300K steps) or 100 epochs, whichever comes first. To get the best performance, we optimize the initial learning rate and batch size for different finetuning methods based on the 1B LLM via grid search. Finally, we set the learning rate to $3e^{-1}, 1e^{-2}$ and $1e^{-3}$ for Prompt, LoRA and FMT, respectively, and set the batch size to 16 and 128 for PET and FMT, respectively.

Table 4: Fitted scaling parameters for different settings.

| Params | WMT14 En-De | | | WMT19 En-Zh | | | MLSum | | |
|---|---|---|---|---|---|---|---|---|---|
| | FMT | Prompt | LoRA | FMT | Prompt | LoRA | FMT | Prompt | LoRA |
| Scaling for LLM model size and finetuning data size | | | | | | | | | |
| $A_m$ | $1.2 \times 10^5$ | $3.9 \times 10^3$ | $2.1 \times 10^3$ | $3.3 \times 10^3$ | $8.5 \times 10^2$ | $6.6 \times 10^2$ | $3.3 \times 10^2$ | 23 | 26 |
| $\alpha_m$ | 0.52 | 0.4 | 0.36 | 0.34 | 0.33 | 0.31 | 0.24 | 0.1 | 0.11 |
| Scaling for pretraining data size and finetuning data size | | | | | | | | | |
| $A_p$ | $6.3 \times 10^2$ | $2.7 \times 10^2$ | $1.4 \times 10^2$ | $2.4 \times 10^2$ | $2 \times 10^2$ | $1.3 \times 10^2$ | 42 | 16 | 17 |
| $\alpha_p$ | 0.21 | 0.21 | 0.18 | 0.17 | 0.2 | 0.18 | 0.11 | 0.069 | 0.073 |
| Scaling for PET parameter size and finetuning data size | | | | | | | | | |
| $A_t$ | - | 1 | 1.4 | - | 1 | 1.2 | - | 2.6 | 2.4 |
| $\alpha_t$ | - | 0.0027 | $-0.0017$ | - | 0.0019 | 0.0044 | - | 0.0026 | 0.00022 |
| $E$ | 0.75 | 0.62 | 0.62 | 1 | 0.77 | 0.73 | 0.98 | 0.00051 | 0.2 |
| $\beta$ | 0.15 | 0.051 | 0.081 | 0.14 | 0.015 | 0.025 | 0.087 | 0.025 | 0.03 |

Figure 8: Generation quality (BLEURT/RougeL) for scaling **pretraining data size and finetuning data size** on WMT14 En-De, WMT19 En-Zh and MLSum.

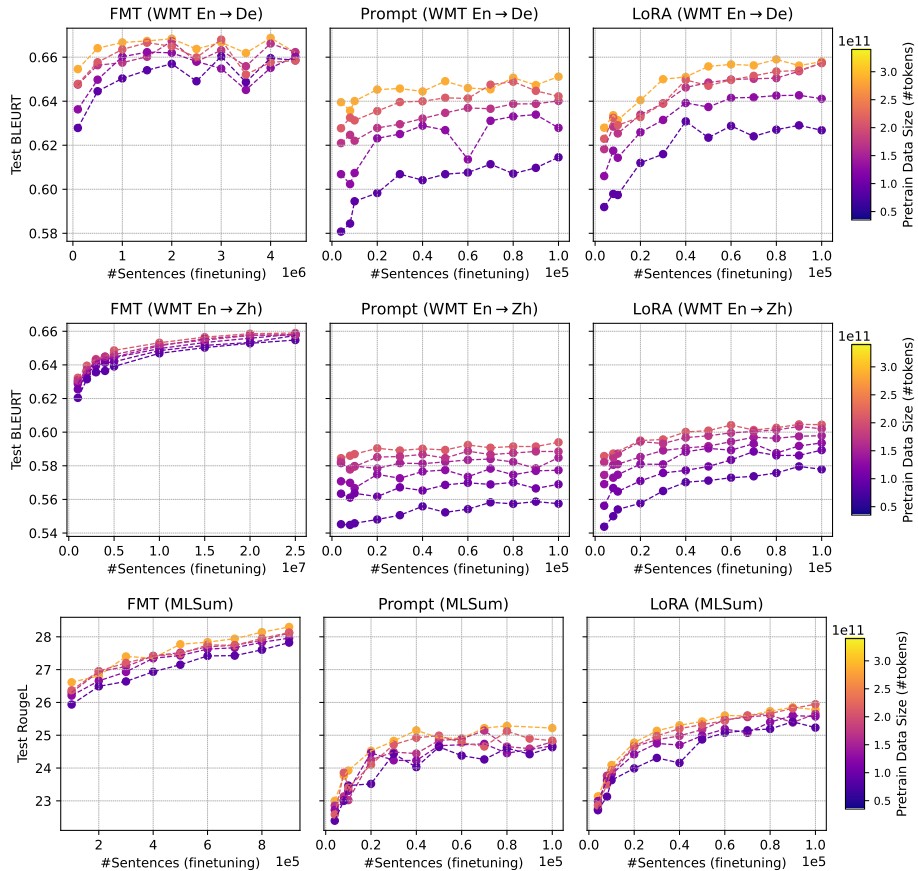

**Analyzing the critical finetuning data size $D_f^c$.** While Eq. (1) hinders us from computing $D_f^c$ directly, it still allows for theoretical analysis between two finetuning methods when their performance gap is a constant:

$$\hat{\mathcal{L}}_1 - \hat{\mathcal{L}}_2 = E_1 - E_2 \implies \hat{D}_f^c = H * X^\gamma, \quad H = \left( A_1/A_2 \right)^{\frac{1}{\beta_1 - \beta_2}}, \gamma = \frac{\alpha_2 - \alpha_1}{\beta_1 - \beta_2} \quad (3)$$

, which follows another power-law. Intuitively, the exponent $\gamma$ captures the transferability difference of the two methods to the downstream task as scaling factor $X$. We summarize the coefficients for

Figure 9: Generation quality (BLEURT/RougeL) for scaling **PET parameter size and finetuning data size** on WMT14 En-De, WMT19 En-Zh and MLSum.

Table 5: Coefficients in Eq. (3) by comparing different methods over setups. "F/P/L": FMT/Prompt/LoRA.

| Params | WMT14 En-De | | | WMT19 En-Zh | | | MLSum | | |
|---|---|---|---|---|---|---|---|---|---|
| | F vs. P | F vs. L | P vs. L | F vs. P | F vs. L | P vs. L | F vs. P | F vs. L | P vs. L |
| Scaling LLM model size and finetuning data size | | | | | | | | | |
| $H$ | $3.7 \times 10^{14}$ | $2 \times 10^{24}$ | $1.6 \times 10^{-9}$ | $6.1 \times 10^{4}$ | $1.6 \times 10^{6}$ | $1.8 \times 10^{-11}$ | $3.6 \times 10^{18}$ | $1.2 \times 10^{17}$ | $0.00045$ |
| $\gamma$ | $-1.2$ | $-2.4$ | $1.5$ | $-0.12$ | $-0.3$ | $1.9$ | $-2.1$ | $-1.8$ | $2.3$ |
| Scaling pretraining data size and finetuning data size | | | | | | | | | |
| $H$ | $3.7 \times 10^{3}$ | $6.9 \times 10^{8}$ | $7.7 \times 10^{-10}$ | $5$ | $2.7 \times 10^{2}$ | $8.6 \times 10^{-19}$ | $3.7 \times 10^{6}$ | $1 \times 10^{7}$ | $1.6 \times 10^{2}$ |
| $\gamma$ | $-0.0015$ | $-0.5$ | $1.2$ | $0.26$ | $0.093$ | $2.1$ | $-0.63$ | $-0.63$ | $-0.63$ |

different tasks in Table 5, where the value differs greatly over tasks and there are no clear patterns across settings.

Figure 10: Fitted single-variable scaling laws for En-De and En-Zh **LLM pretraining**. We evaluate the model on a held-out validation set and fit the scaling law based on PPL. Note that the scaling law doesn't well extrapolate to 16B for En-Zh LLM whose actual performance is worse than the expectation (This might be caused by pretraining instabilities.). Such mismatch we argue is amplified after finetuning as shown in Figure 2.

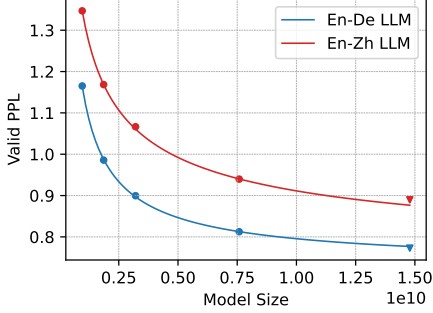

**How does finetuning affect the few-shot capability of the base LLM?** Apart from zero-shot translation, we also explore LLM's few-shot capability after finetuning. Few-shot generation not

Table 6: Held-out fitting errors (↓) for the additive and multiplicative scaling formulation over different tasks. Overall, multiplicative scaling law generalizes better.

| | Scaling Factor | Multiplicative | | | | Additive | | | |
|---|---|---|---|---|---|---|---|---|---|
| | | FMT | Prompt | LoRA | Avg | FMT | Prompt | LoRA | Avg |
| WMT En-De | LLM Model Size | 0.0052 | 0.0043 | 0.0047 | **0.0048** | 0.012 | 0.0076 | 0.0045 | 0.0079 |
| | Pretraining Data Size | 0.0057 | 0.0061 | 0.0084 | **0.0068** | 0.0048 | 0.0075 | 0.0082 | 0.0069 |
| | PET parameter size | - | 0.005 | 0.0031 | **0.004** | - | 0.0069 | 0.0032 | 0.005 |
| WMT En-Zh | LLM Model Size | 0.0075 | 0.019 | 0.026 | **0.018** | 0.021 | 0.018 | 0.029 | 0.022 |
| | Pretraining Data Size | 0.002 | 0.0071 | 0.0056 | **0.0049** | 0.0026 | 0.0069 | 0.0058 | 0.0051 |
| | PET parameter size | - | 0.0075 | 0.0051 | 0.0063 | - | 0.0076 | 0.0044 | **0.006** |
| MLSum | LLM Model Size | 0.0066 | 0.013 | 0.022 | 0.014 | 0.0072 | 0.015 | 0.017 | **0.013** |
| | Pretraining Data Size | 0.009 | 0.0083 | 0.0039 | 0.007 | 0.0062 | 0.0046 | 0.0043 | **0.005** |
| | PET parameter size | - | 0.0081 | 0.003 | 0.0055 | - | 0.0053 | 0.0027 | **0.004** |

Table 7: Pearson correlation between PPL and BLEURT/RougeL for different finetuning methods and setups. "‡": the correlation is significant at $p < 0.01$. Note lower PPL and higher BLEURT/RougeL denote better quality, thus their correlation values are negative. In general, PPL and BLEURT/RougeL are highly correlated.

| | Scaling Factor | FMT | Prompt | LoRA |
|---|---|---|---|---|
| WMT En-De | LLM Model Size | -0.184 | -0.986[‡] | -0.988[‡] |
| | Pretraining Data Size | -0.792[‡] | -0.967[‡] | -0.980[‡] |
| | PET parameter size | - | -0.841[‡] | -0.975[‡] |
| WMT En-Zh | LLM Model Size | -0.984[‡] | -0.994[‡] | -0.995[‡] |
| | Pretraining Data Size | -0.994[‡] | -0.979[‡] | -0.978[‡] |
| | PET parameter size | - | -0.643[‡] | -0.968[‡] |
| MLSum | LLM Model Size | -0.965[‡] | -0.909[‡] | -0.890[‡] |
| | Pretraining Data Size | -0.941[‡] | -0.833[‡] | -0.838[‡] |
| | PET parameter size | - | -0.924[‡] | -0.986[‡] |

only offers a way to inspect LLM's capability but also is of interest to downstream applications as it provides an effective way to adapt models over domains. Figures 11, 12, 13 and 14 shows the impact of finetuning on few-shot generation.

We note that FMT degenerates LLM's few-shot capability in most cases, where adding more finetuning data often reduces the few-shot performance. By contrast, PET behaves more robustly which retains most of LLM's few-shot capability regardless of model size and pretraining data size.

Figure 11: One-shot performance (BLEURT/RougeL) for LLM model size and finetuning data size scaling on WMT14 En-De, WMT19 En-Zh and MLSum. 'Baseline': performance without finetuning.

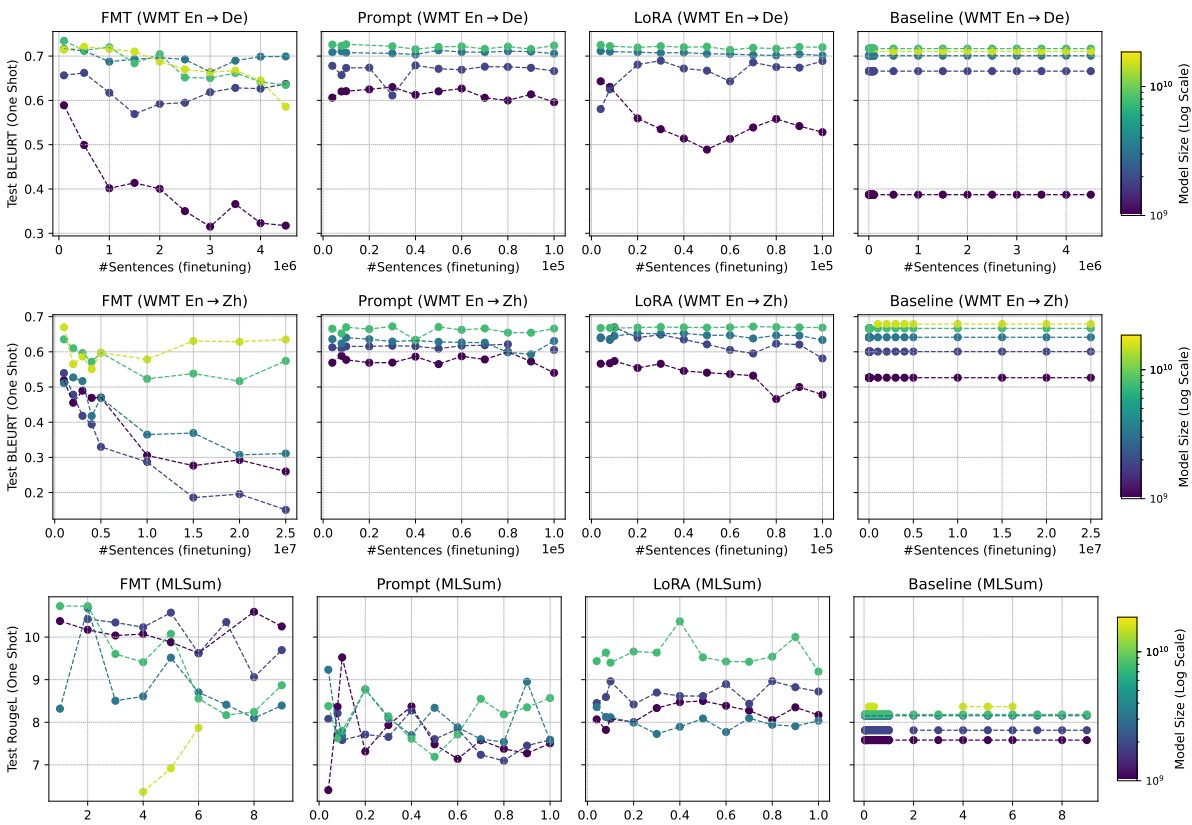

Figure 12: Five-shot performance (BLEURT/RougeL) for LLM model size and finetuning data size scaling on WMT14 En-De and WMT19 En-Zh.

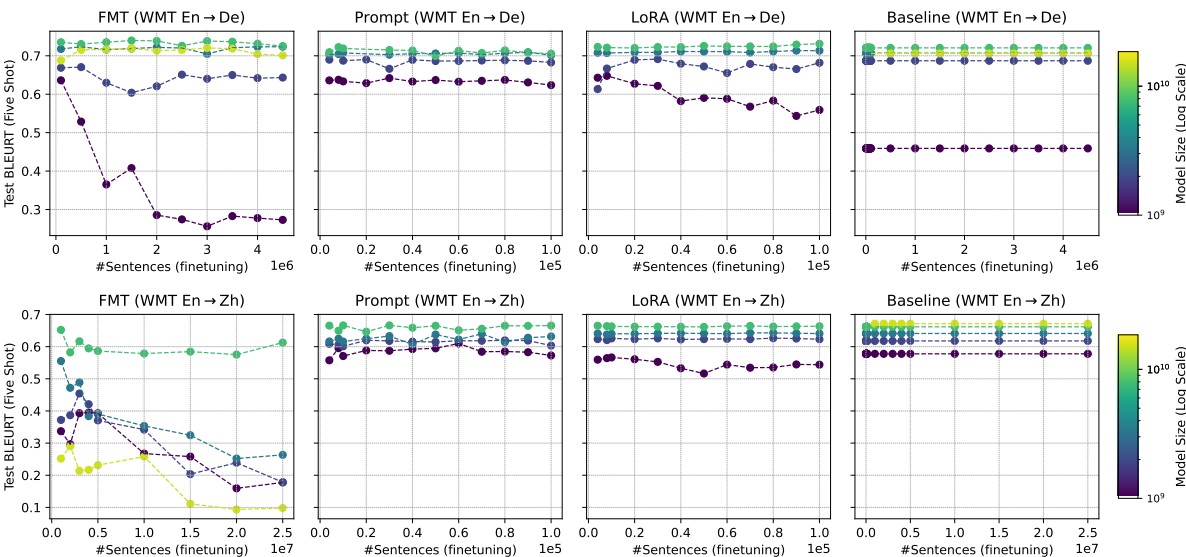

Figure 13: One-shot performance (BLEURT/RougeL) for pretraining and finetuning data size scaling on WMT14 En-De, WMT19 En-Zh and MLSum. 'Baseline': performance without finetuning.

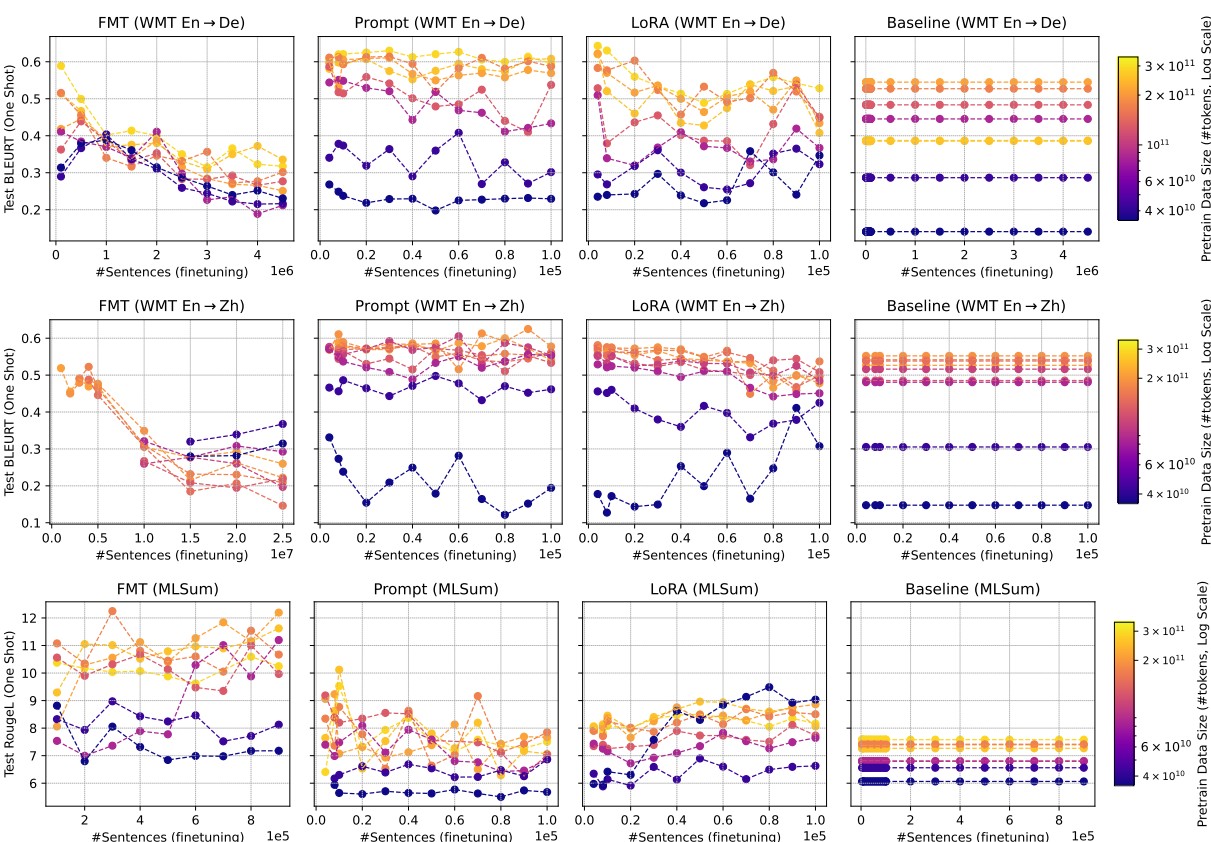

Figure 14: Five-shot performance (BLEURT/RougeL) for pretraining and finetuning data size scaling on WMT14 En-De and WMT19 En-Zh.

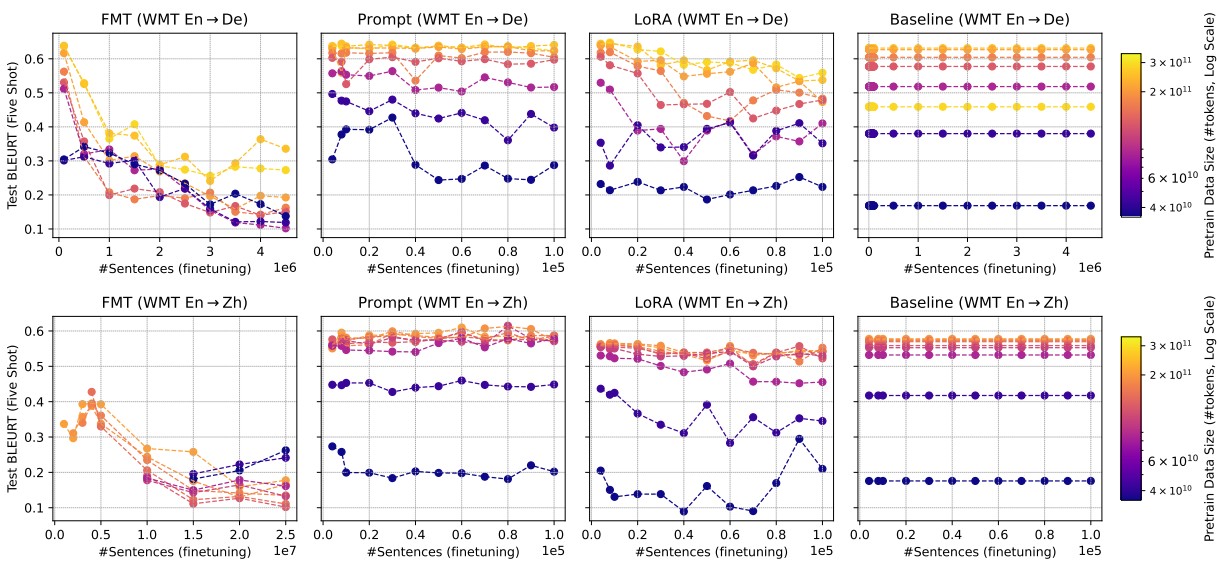

