# OpenReview forum: "When Scaling Meets LLM Finetuning: The Effect of Data, Model and Finetuning Method"
_ICLR.cc/2024/Conference — ICLR 2024 poster_

### Official Review · Reviewer_oDvP · 2023-10-29

**Soundness:** 4 excellent
**Presentation:** 4 excellent
**Contribution:** 3 good
**Rating:** 8
**Confidence:** 3

**Summary:**

This paper examines scaling laws for LLM adaptation methods (finetuning, PET like LoRA and PT), where the LLMs were trained on two different machine translation corpora (En->ZH and EN->De) and adapted to either the other translation task, or article summarization in an unseen language.
The paper is primarily empirical, and the goal is to fit scaling laws and identify general trends that could transfer across problem domains (e.g. vision, multimodal, etc).

**Strengths:**

### Overall
The paper is clearly written. The methodology is generally explicit, and the authors do a good job of interpreting the (many) experiments in the paper.

The figures are clear and well-organized.

Excellent references to existing work in identifying scaling laws.

### Experiments
The experimental results are somewhat expected (given enough data, use FMT, with limited data, use PT or LoRA). The authors do a good job of explaining and justifying these conclusions given experimental evidence, with the appropriate amount of uncertainty given the noisiness of the data. I think that correctly calibrating the uncertainty is a great strength of the paper.

Excellent analysis of scaling different axes (pretraining data, model size, finetuning data)

**Weaknesses:**

### Experiments
#### Fig 2:
Given the poor extrapolation to the 16B model, I'd like to understand the cause -- i.e. is this the right form of the equation, or does this multiplicative scaling break down at larger parameter counts?
One way to test this might be to compare the fit by including 16B in the fitting procedure, vs excluding it.

#### Fig 4:
I liked the analysis of scaling to larger PET settings (LoRA and PT in Fig 4). Since the datasets seem to support up to 1e-6 and 1e-7 training sentences, I wish the authors had compared the scaling on even larger finetuning datasets to enable direct comparison to FMT in Figure 3.

#### Fig 5
I didn't really understand Figure 5. My my understanding is that the figure analyzes the fitted power laws, and the x-y- point indicates the estimated amount of finetuning necessary to achieve the performance parity between the two approaches at different model sizes. Does the ordering here matter (e.g. FMT vs LoRA, or LoRA vs FMT) -- my understanding is that it should not.

**Questions:**

- How does zero-shot evaluation work in Figure 7? If the source language is unseen, how is the model able to get nontrivial performance on the task?

### Typos:
- P4 footnote: is pretty week → pretty weak
- Fig 2. center column: Propmt → Prompt

---

> ### Author Response · Authors · 2023-11-20
> **Response to Reviewer oDvP**
>
> Thanks for your insightful comments! We corrected all typos in the updated version.
>
>
> **Re: is this the right form of the equation, or does this multiplicative scaling break down at larger parameter counts?**
>
> Understanding the reason for the bad extrapolation to 16B is valuable. We attempted different scaling formulations (both multiplicative and additive) and found that their extrapolation often fails at 16B mainly on En-Zh LLM. This inspires us that it is something else rather than the formulation causing this problem.
>
> After careful analysis, we discovered that the pretraining performance of the 16B En-Zh LLM can’t be well predicted by single-variable scaling laws, as shown in Figure 10 in the updated version: the actual performance is worse than the expectation. We argue that such mismatch caused by pretraining instabilities gets amplified during finetuning. This issue becomes more severe for PET as PET finetuning suffers more from unstable optimization.
>
>
> **Re: The authors had compared the scaling on even larger finetuning datasets to enable direct comparison to FMT in Figure 3.**
>
> While finetuning LLM with PET on larger-scale datasets (e.g. up to $1e6$~$1e7$ examples) is feasible theoretically, it is non-trivial in practice. This is because the hyperparameters of PET, including the learning rate, batch size and finetuning step, are highly biased and tuned for small-scale datasets to achieve the best quality but they can hardly generalize to larger settings. For example, we set batch size to 16 for PET, with which the model converges very slowly when using $1e6$ examples and may not converge at all due to the used finetuning step constraint. Readjusting these hyperparameters are required but are expensive and it may also break the scaling trend.
>
>
> **Re: Does the ordering here matter (e.g. FMT vs LoRA, or LoRA vs FMT) -- my understanding is that it should not.**
>
> Your understanding is correct: the ordering here doesn’t matter.
>
>
> **Re: How does zero-shot evaluation work in Figure 7? If the source language is unseen, how is the model able to get nontrivial performance on the task?**
>
>
> Although our LLMs are bilingual, they were pretrained on web-scale corpora which often unintentionally include data from other languages. In other words, the model has seen other source languages. We call it “zero-shot evaluation” because we never finetune the model on the “zero-shot” pairs but we expect that LLM is capable of generalizing its translation ability to them.

---

> > ### Comment · Reviewer_oDvP · 2023-11-22
> > **An empirical study**
> >
> > Thank you to the authors, for your replies to myself and the other reviewers. Thanks to the other reviewers for your comments and discussion.
> >
> > I think that low-resource finetuning is a very important practical technique, and one that is poorly understood. This paper is one part of building such an understanding. I think it would make a great ICLR poster.
> >
> > I also think that we should not focus too much on the specific scaling law formulat -- the fit seems rather poor but for now there's no alternative formula. Maybe a better one will come along, and then that hypothetical paper can compare to this one.
> > - To the reviewers: I think we should not penalize the authors if the results turn out not to fit a clean formula -- other authors should get the same results, no?
> > - To the authors: if the trend changes due to small changes in the hyperparameters, then I would ask how robust are these trends? Are the experiments as thorough and the findings as robust as they could be? And finally, if the trend breaks down at larger sizes, then are these really "scaling laws"?

---

> > > ### Comment · Reviewer_gFLk · 2023-11-22
> > > **I agree**
> > >
> > > > To the reviewers: I think we should not penalize the authors if the results turn out not to fit a clean formula -- other authors should get the same results, no?
> > >
> > > I agree. I myself was not doing this. [My comment above](https://openreview.net/forum?id=5HCnKDeTws&noteId=OV3Q64HiwW) says, "The takeaway seems to be that many aspects of fine-tuning are in fact not predictable from a simple scaling law. This is completely fine with me as a discovery about fine-tuning, and the paper offers rich experimental support for it. However, I remain concerned that the paper will be advertised as one that discovers a scaling law for fine-tuning."
> > >
> > > The revised version of the paper takes steps to alleviate my concern about how the paper will be advertised.

---

### Official Review · Reviewer_wU43 · 2023-10-30

**Soundness:** 3 good
**Presentation:** 3 good
**Contribution:** 3 good
**Rating:** 5
**Confidence:** 3

**Summary:**

This paper describe a multiplicative joint scaling law, considering different factors including LLM model size, pretraining data size, finetuning data size, PET parameter size, on machine translation and summarization tasks.

**Strengths:**

- The paper introduces a multiplicative joint scaling law and offers comprehensive experiments to empirically demonstrate that this law applies to both machine translation and summarization.
- This paper provides insightful observations regarding fine-tuning, especially parameter-efficient fine-tuning (PEFT) which I think is the most interesting part for the community now, and its relationship with parameter or data size scaling laws.

**Weaknesses:**

Minor:
-The paper evaluates only MT and summarization. Including more tasks or languages, particularly low-resource translations, would enhance the paper's comprehensiveness.
Major:

Major:
-Regrettably, it seems that the proposed scaling law may exhibit a significant mismatch for parameter-efficient fine-tuning when the model size is 16B. This raises concerns about the law's applicability to larger models, especially those of 70B or exceeding 100B in size.
-The paper omits some details regarding model training. Conventionally, MT models employ an encoder-decoder architecture. I believe all models in this study are decoder-only. How did the authors approach training with a decoder-only architecture for MT tasks? How might this differ from the scaling laws when using an encoder-decoder model? What prompts were utilized for MT training?

**Questions:**

Please see weaknesses above.

---

> ### Author Response · Authors · 2023-11-20
> **Response to Reviewer wU43**
>
> Thanks for your insightful comments!
>
>
> **Re: Including more tasks or languages, particularly low-resource translations, would enhance the paper's comprehensiveness.**
>
>
> We agree that working with different tasks and/or languages could enhance the generality and comprehensiveness of our study. That’s exactly why we performed experiments on two different language pairs (En-De and En-Zh) for machine translation as well as multilingual summarization. Adding more tasks should be beneficial, which we leave to the future given the timeline and compute resource constraints.
>
>
> **Re: it seems that the proposed scaling law may exhibit a significant mismatch for parameter-efficient fine-tuning when the model size is 16B.**
>
>
> When extrapolating to 16B, we noticed a high mismatch particularly for LoRA and Prompt on En-Zh LLM. After careful analysis, we argue that this problem stems more from the pretraining and finetuning rather than the proposed scaling law. Specifically, we find that the pretraining performance of the 16B En-Zh LLM is not well predicted according to a single-variable scaling law as shown in Figure 10 in the updated version. The 16B model performs slightly worse than the expectation, which we argue is caused by pretraining instabilities (such as job corruptions). Such mismatch is further amplified by the finetuning as PET finetuning suffers more from unstable optimization.
>
>
> **Re: How did the authors approach training with a decoder-only architecture for MT tasks? What prompts were utilized for MT training?**
>
> We perform finetuning for MT by converting MT examples into the LLM format and optimize the model with the conditional log-likelihood objective. More concretely, for each *source* and *target* example in MT, we concatenate them into a single sequence, i.e. *source* <sep> *target*. We feed this sequence to LLM and only compute log-likelihood on the *target*. We added this in the Appendix in the updated version.
>
>
> **Re: How might this differ from the scaling laws when using an encoder-decoder model?**
>
> Based on prior studies, the scaling of encoder-decoder models shows both similarities and differences compared to that of decoder-only models [1][2]. Exploring the scaling of finetuning encoder-decoder models is definitely an interesting research problem, which we leave to the future.
>
> >- [1] Ghorbani et al., 2021. Scaling  laws  for  neural  machine  translation.
> >- [2] Bansal et al., 2022. Data  scaling  laws  in  NMT:  The  effect  of  noise  and  architecture.

---

> > ### Comment · Reviewer_wU43 · 2023-11-22
> >
> > Thanks to the authors for their reply. I would stand for my score.

---

### Official Review · Reviewer_R4DF · 2023-10-31

**Soundness:** 3 good
**Presentation:** 3 good
**Contribution:** 3 good
**Rating:** 8
**Confidence:** 4

**Summary:**

The work provides a set of simple but extensive scaling law experiments on comparing pretraining, fine-tuning, and parameter-efficient tuning LLMs on translation and summarization tasks.

**Strengths:**

- This work provides a set of straightforward and well-motivated set of experiments.  The authors are clear where there are reasonable scaling patterns, and cases where there is no discernible pattern.

**Weaknesses:**

- Some typographical/grammar mistakes. E.g. "propmt" in Figure 2, "infers" rather than "implies", "there exits a critical point" etc
- It is unclear to me if the set of tasks chosen (translation, and multi-lingual summarization) are representative of broader applications of fine-tuning. However, I think the results stand on their own for this set of narrow applications at least.

**Questions:**

- What model sizes are used in Figures 2/3/4?
- Figure 5 is unclear to me. Does the critical point refer to when A outperforms B in "A vs B"?

---

> ### Author Response · Authors · 2023-11-20
> **Response to Reviewer R4DF**
>
> Thanks for your insightful comments! We corrected all typos and grammatical errors in the updated version.
>
>
> **Re: What model sizes are used in Figures 2/3/4?**
>
>
> In Figure 2, we used LLMs of parameters from 1B to 16B. In Figures 3 and 4, we mainly used 1B LLM. We added this information in the updated version.
>
>
> **Re: Figure 5 is unclear to me. Does the critical point refer to when A outperforms B in "A vs B"?**
>
>
> The critical point for “A vs. B” indicates the exact finetuning data size at which A performs equal to B given the different base model setup (x-axis) like model size or pretraining data size. We also explained this in the updated version.
>
> For example, if we investigate how the pretraining data size of the base 1B LLM would affect the decisions on finetuning data size, we can conclude from the plot (bottom-left) that when the pretraining data size is 2e11, the critical point for FMT vs LoRA at 50K indicates that we're more likely to achieve better performance with LoRA when we have data less than 50K while we have more headroom with FMT when we have more than 50K data.

---

### Official Review · Reviewer_gFLk · 2023-11-01

**Soundness:** 2 fair
**Presentation:** 3 good
**Contribution:** 3 good
**Rating:** 6
**Confidence:** 3

**Summary:**

This paper proposes a scaling law for language model fine-tuning. The claim is supported by experiments with fine-tuning in machine translation and summarization tasks. The other scaling factors besides fine-tuning dataset size are LLM size, pretraining data size, and PET parameter sizes for the Prompt Tuning and LoRa fine-tuning regimes. The core finding is that a multiplicative scaling model achieves better fits than a closely additive scaling model.

**Strengths:**

It's excellent to see scaling law investigations extended to fine-tuning, where they can potentially provide a lot of practical value.

The paper is generally clear and direct, and the experimental findings are quite rich.

**Weaknesses:**

The precise nature and strength of the findings is difficult for me to discern, and I am not sure of the value of this result for fine-tuning work. All of this leaves me concerned about the paper, but I am open-minded. I am going to express my concerns as questions and see what the answers are.

**Questions:**

1. Table 2 shows that the multiplicative scaling is better than the additive one for WMT14 En-De. However, the multiplicative model seems strictly more expressive than the additive one, so I am not sure this is surprising. Why not add even more terms for even more expressivity? Can there be, and should there be, some controlling for the complexity of the law itself?

2. Where are the counterparts of Table 2 for the other tasks and other metrics (besides perplexity)? My apologies if I am overlooking something in the paper. It seems like these other numbers would be given prominently.

3. The paper does say that the BLEURT-RougeL picture "shows high correlation with the PPL scores in general", but eye-balling Figure 7 in the appendix doesn't support this too well, though it's easy to imagine that the quantitative picture is different. But what is the quantitative picture?

4. The promise of the paper is that the scaling law will provide guidance for people seeking to fine-tune. However, the guidance seems to me that perplexity will generally go down for all methods, but that the best method and precisely ideal stopping point will be highly variable. This guidance is very familiar and doesn't need to be characterized with a "scaling law". Is there more specific guidance implicit in this work?

__The authors gave thoughtful answers to the above questions, which helped me understand the work better, and I raised my score by 1 point in response.__

---

> ### Author Response · Authors · 2023-11-20
> **Response to Reviewer gFLk**
>
> Thanks for your insightful comments!
>
>
> **Re: The core finding is that a multiplicative scaling model achieves better fits than a closely additive scaling model.**
>
> We appreciate your highlight of the proposed multiplicative scaling law. Note we consider it as a tool for analyzing the empirical results. Our main focus is to understand how LLM model size, pretraining data size, PET parameter size and finetuning data size affect the finetuning, which we achieve through this tool. We hope our follow-up findings could gain more attention, such as the importance of LLM model scaling, the ineffectiveness of PET scaling, and the high degree of task dependence of LLM finetuning.
>
>
> **Re: Why not add even more terms for even more expressivity? Can there be, and should there be, some controlling for the complexity of the law itself?**
>
> Expressivity is not the only consideration and we expect the law to be as simple/clean as possible.
> 1) There should be a trade off between expressivity and generalization. Adding more terms increases the expressivity theoretically but also brings in risks of overfitting, i.e. the model fits well on the given data but suffers from poor extrapolation. This problem becomes more concerning in LLM finetuning as the finetuning results can be very noisy, particularly for LoRA and Prompt tuning.
> 2) Keeping the formulation simple eases analysis. Note our analysis in Section 4 is mainly based on the comparison of different scaling exponents. But such direct comparison is significant only for the proposed multiplicative formulation without additional terms. In the additive formulation, for example, comparing $\alpha$ with $\beta$ is complicated due to the difference of $A$ and $B$.
>
>
> **Re: Where are the counterparts of Table 2 for the other tasks and other metrics (besides perplexity)?**
>
> For the fitting errors on WMT En-Zh and MLSum, we add them in Table 6, Appendix in the updated version. We noticed that the multiplicative scaling law performs worse on MLSum compared to the additive one; but overall, our proposal generalizes slightly better over different tasks and finetuning methods.
>
>
> For the scaling laws on metrics other than PPL, we didn’t explore them. Firstly, these metrics highly correlate with PPL as demonstrated by the high Pearson’s $r$ in Table 7, Appendix in the updated version. Secondly, adopting PPL for scaling law is the standard practice in the literature [1,2,3] because improvements in PPL may not be well reflected in other metrics. Take the binary classification task as an example. Improving the probability (thus PPL) of the correct answer from 0.1 to 0.2 doesn’t change the classification accuracy (take 0.5 as the threshold).
>
>
> **Re: BLEURT-RougeL picture "shows high correlation with the PPL scores in general"**
>
> We performed Pearson correlation analysis between PPL and BLEURT/RougeL and added Table 7 to Appendix in the updated version. Most Pearson’s $r$’s absolute value is larger than 0.9, and all of them are significant at $p<0.01$ except FMT for LLM model scaling on WMT En-De.
>
>
> **Re: This guidance is very familiar and doesn't need to be characterized with a "scaling law". Is there more specific guidance implicit in this work?**
>
> While the findings align with intuition and look familiar, the scaling law is still necessary as it offers a quantitative measure for the impact of different scaling factors on the finetuning. This enables the comparison between scaling factors and gives readers more convincing evidence beyond intuition and experience. We consider these findings reached through large-scale and systematic experiments and characterized by our scaling laws as significant contributions to the community.
>
>
> We need to emphasize that our results provide rich information and the guidance is multi-dimensional, not just specific to finetuning. For example, finetuning benefits more from LLM model scaling than pretraining data scaling, which suggests a different recipe compared to the computation optimal scaling for the pretraining. In terms of finetuning, while PPL generally goes down with more data, the expected return from collecting more finetuning data differs greatly across finetuning methods as demonstrated by the scaling laws, and PET shouldn’t be expected to perform well with small LLMs.
>
>
> In the era of LLM, our study represents itself as the first kind to explore the model, data and finetuning method for the scaling of LLM finetuning, which we believe contains intriguing and substantial contents to the community. We hope the reviewer could reconsider our paper.
>
>
> >- [1] Kaplan et al., 2020. Scaling Laws for Neural Language Models
> >- [2] Ghorbani et al., 2021.  Scaling Laws for Neural Machine Translation
> >- [3] Hoffmann et al., 2022. Training Compute-Optimal Large Language Models

---

> > ### Comment · Reviewer_gFLk · 2023-11-21
> > **"Scaling law" framing distracts from the experimental findings**
> >
> > I appreciate the authors' responses and additions to the paper. I am raising my score by 1 point. Based on the discussion in this forum, I feel it's a mistake to say the paper has discovered a "scaling law". The scaling law looks more like a sort of guiding hypothesis that turns out to be kind of true but also pretty much wrong in many relevant scenarios. The takeaway seems to be that many aspects of fine-tuning are in fact not predictable from a simple scaling law. This is completely fine with me as a discovery about fine-tuning, and the paper offers rich experimental support for it. However, I remain concerned that the paper will be advertised as one that discovers a scaling law for fine-tuning.

---

> > > ### Author Response · Authors · 2023-11-22
> > > **Thanks for your kind comments**
> > >
> > > As we explained in the above response, our paper is mainly for understanding LLM finetuning and we regard the scaling law just as a tool for the analysis. That's why we didn't include the term "scaling law" in the title. To avoid concerns like yours, we further replaced the term "establish" with "propose" for the scaling law in the updated version.
> > >
> > > We also acknowledge that the scaling laws are imperfect in several aspects. Note we explicitly stated in Section 7 that *The  proposed joint  scaling  law is  mostly  based  on  empirical  results  on  closed  generation  tasks  without  theoretical  groundings.*

---

### Meta-Review · Area_Chair_bxyf · 2023-12-13

**Metareview:**

The paper conducts an empirical analysis of fine-tuning LLMs for downstream machine translation and summarization tasks in order to establish scaling laws describing how various dimensions of pre-training and fine-tuning interact to determine final performance. Reviews are mostly in favor of acceptance, and all reviews point out the value of establishing scaling laws for fine-tuning -- an empirical analysis currently missing from the literature. Reviewers did identify several weaknesses. First, unlike established scaling laws for pre-training, which are consistent and accurate, the results in this paper are complex and somewhat noisy. This led reviewers to question whether it is accurate to claim a scaling law has been established for fine-tuning, given that the results don't seem to closely follow a single general scaling law. Second, some reviewers pointed out that studying machine translation and summarization alone may not lead to scaling laws that generalize to other fine-tuning tasks. Nonetheless, reviewers agree that these results are worth knowing about -- even if a clear and general scaling law cannot currently be established.

**Justification For Why Not Higher Score:**

The results in this paper don't establish a clear and general scaling law. Results are limited to two fine-tuning tasks: translation and summarization.

**Justification For Why Not Lower Score:**

The results of this empirical analysis are worth knowing about. This kind of study is currently missing from the literature.

---

### Decision · Program_Chairs · 2024-01-16

Accept (poster)